# First Record of the Red Cornetfish *Fistularia petimba* Lacepède, 1803 from Amorgos Island (Central Aegean Sea; Greece) and a Review of Its Current Distribution in the Mediterranean Sea

Nefeli Tsaousi [ID] and Stefanos Kalogirou *[ID]

Laboratory of Applied Hydrobiology, Department of Animal Science, School of Animal Sciences, Agricultural University of Athens, Iera Odos 75, 11855 Athens, Greece; tsaousinefeli@gmail.com
* Correspondence: stefanos.kalogirou@aua.gr; Tel.: +30-2105294459

**Abstract:** The rapid spread of non-native species (NNS) poses a significant threat to biodiversity globally, with the Mediterranean region being particularly susceptible due to increased human activities and its status as a marine biodiversity hotspot. In this study, we focus on the introduction and distribution of *Fistularia petimba*, a member of the Fistulariidae family, in the eastern Mediterranean Sea and a record from the coasts of Amorgos Island, Greece. Through a baseline fishery study conducted over 12 months, utilizing experimental sampling with gillnets, trammel nets, and longlines, one individual of *F. petimba* was captured off the coast of Katapola Bay. Morphological examination confirmed its identity, with measurements on meristic characteristics obtained and the stomach content analysed. This finding represents a significant addition to the documented distribution of *F. petimba* in the Mediterranean Sea, particularly in the Aegean Sea, underscoring the importance of ongoing research in uncovering new occurrences and expanding our understanding of marine biodiversity and ecosystem changes. Further investigation into the ecological preferences and population dynamics of *F. petimba* in the Aegean Sea is crucial for informed conservation and management efforts if this species is considered to be established.

**Keywords:** *Fistularia petimba*; red cornetfish; Mediterranean Sea; invasive species; non-native species; Lessepsian species; marine biodiversity; species distribution

**Key Contribution:** This study provides a review of a recent non-native species geographical distribution. This study provides insights on the crucial establishment phase for the species and signifies areas for further investigation to reveal life trait characteristics and succession rate.

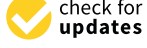



## 1. Introduction

One of the main threats that biodiversity currently faces is the rapid spread of non-native species (NNS). NNS are defined as an array of species spreading outside their natural or native distribution range [1]. Different areas worldwide have been experiencing vast impacts from the introduction of NNS, often related to increased human activities, such as the opening of canals, the continuous growth of the shipping industry across biogeographic barriers [2,3], a wide range of changes in water temperature due to climate change [4–6], fishing pressure [7–9], and habitat degradation or loss of species [10–12]. In the studied area of the Mediterranean Sea, recognized as one of the main hotspots of marine biodiversity [13,14], the effects of NNS are apparent, both in terms of introduction rate [15] and number of introduced species [16], leading to the global acknowledgment of the Mediterranean region as a hotspot area for NNS [17].

A region where visible changes in aquatic biodiversity have occurred is the eastern Mediterranean Sea, where a rapid introduction of fish species of Indo-Pacific origin are observed, i.e., the Levantine Sea [18–21], significantly raising the overall amount of fish biomass up to 90% in specific habitats [22,23]. These habitats include hard bottoms

for *Siganus luridus* and *Siganus rivulatus*, and sandy bottoms and seagrass meadows for the *Lagocephalus sceleratus* [22]. Though Indo-Pacific fish species could potentially arrive through various ways in the Levantine Basin, they most likely arrive through immigration via the Suez Canal, which opened to shorten the commercial shipping ways between the Indian Ocean and the Mediterranean Sea in 1869 [24]. It is assumed that species that normally resided in the Red Sea and the Indian Ocean traversed through the Suez Canal and proceeded northwards along the Levant coast, actively or passively aided by human activity [25]. These species were named Lessepsian after the name of the constructor of the canal, engineer, and diplomat Ferdinand de Lesseps [26].

The aforementioned group of Lessepsian species established in the Mediterranean Sea currently includes *Fistularia commersonii* [27] and *Fistularia petimba*, also called cornetfishes or flutemouths, which belong to the Fistulariidae (order of Syngnathiformes). There is only one genus in this family, *Fistularia*, and four different species: *Fistularia commersonii* Rüppell, 1838; *Fistularia corneta* Gilbert and Starks, 1904; *Fistularia petimba* Lacepède, 1803; and *Fistularia tabacaria* Linnaeus, 1758 [28]. The species *F. tabacaria* inhabits the tropical Atlantic, while its closest relative *F. commersonii* inhabits the Pacific and Indian Oceans. *Fistularia petimba* spans the tropical Atlantic and Indo-West Pacific Oceans, whereas *F. corneta* is confined to the tropical eastern Pacific [29]. Fistularidae species are predators, inhabiting shallow waters of tropical and subtropical areas [29]. Although *F. commersonii* originated from the Indo-Pacific region [30], a wide geographical distribution has been observed in the eastern Mediterranean Sea [31], with multiple sightings of this Lessepsian immigrant. Due to its rapid growth and reproduction cycle, it has successfully formed large populations in the areas where it has been established, with notable ecological impacts on the native species [32]. *Fistularia commersonii* is a piscivorous species, mainly feeding on smaller fish and complementing its diet with some Crustacea species. As the size of the species increases, a corresponding increase in the size of prey consumed has been found [32].

The studied red cornetfish, *F. petimba*, is native to the Indo-West Pacific, the tropical Atlantic [30], and the East Atlantic Ocean [33]. With a time lag of 20 years since its first siting in the western Mediterranean Sea (1996), it has been reported in several locations in the eastern Mediterranean Sea over the last ten years [34,35]. In this study, we show the immigration path of *F. petimba* in the eastern Mediterranean Sea through a stepping stone process of establishment through the Suez Canal [36,37].

## 2. Materials and Methods

### 2.1. Sampling Methodology

A monthly experimental fishery sampling was performed in Amorgos Island, Central Aegean Sea, between September 2022 and August 2023 (Figure 1).

The sampling method used involved three types of gears: gillnets (GNSs), trammel nets (GTRs), and long lines (LLSs). The gears used were designed to study the population dynamics of targeted fisheries species.

For GNSs and GTRs, nine different mesh sizes (20, 22, 24, 26, 28, 30, 32, 36, 45 in mm) were used, and for LLS six different hook sizes (9, 10, 11, 12, 13, 14) were used to reflect the most commonly used mesh and hook sizes in small-scale fisheries of the Aegean Sea.

The nets used in this study had a total length of 900 m, consisting of 100-m panels for each of the nine mesh sizes. The height of each compartment was 100 meshes. The arrangement of mesh sizes for both GNSs and GTRs were randomly selected and are illustrated in Figures 2 and 3. For longlines, each compartment had a length of 200 m and was equipped with 200 hooks (100 hooks per hook size). One compartment comprised of hook sizes 9 and 10, and two more compartments comprised of combinations of hook sizes 11–12 and 13–14, respectively.

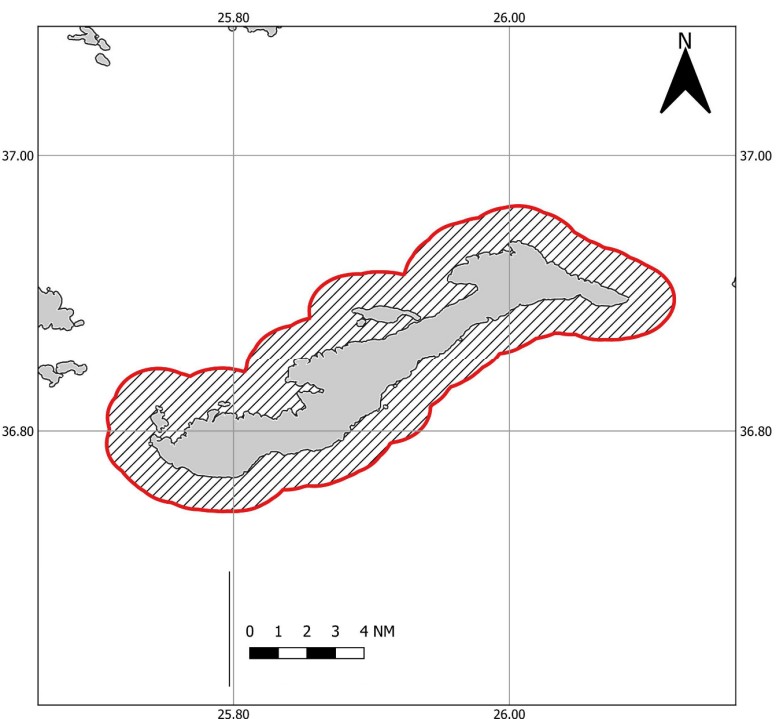

**Figure 1.** Map of the Amorgos Island, Central Aegean Sea, Greece. The red line marks the sampling area around Amorgos Island.

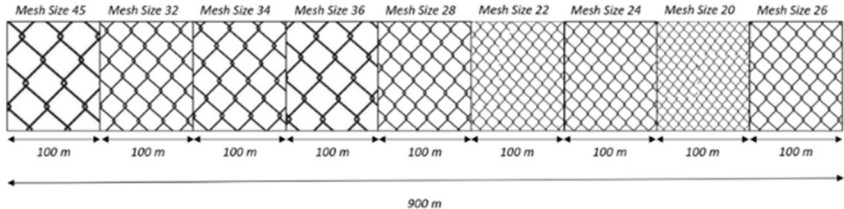

**Figure 2.** GNS mesh size arrangement in this study.

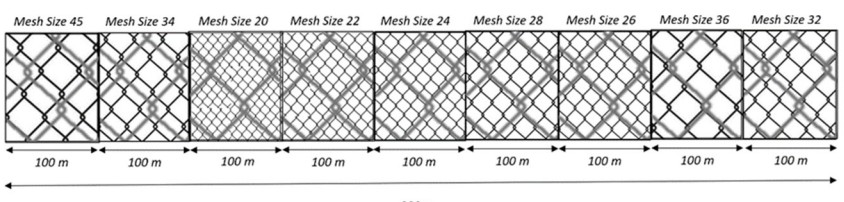

**Figure 3.** GTR mesh size arrangement in this study.

All species samples were stored in a freezer ($-20\ °C$) until transportation and were deposited to the Laboratory of Applied Hydrobiology of the Agricultural University of Athens, Greece, for further examination.

### 2.2. Identification of the Species

*Fistularia petimba* was identified based on its morphological characteristics [29], following the genus description given by Fritzsche (1976): *Fistularia* species can be identified by their elongated body.

*Fistularia petimba* was distinguished from its confamilial species by its specific morphological features [29]: number of rays on the dorsal fin (13–17) and the anal fin (13–16), elongated bony plates embedded in the skin along the midline of the back, with posterior lateral line ossifications terminating in a retrorse spine (Figure 4).

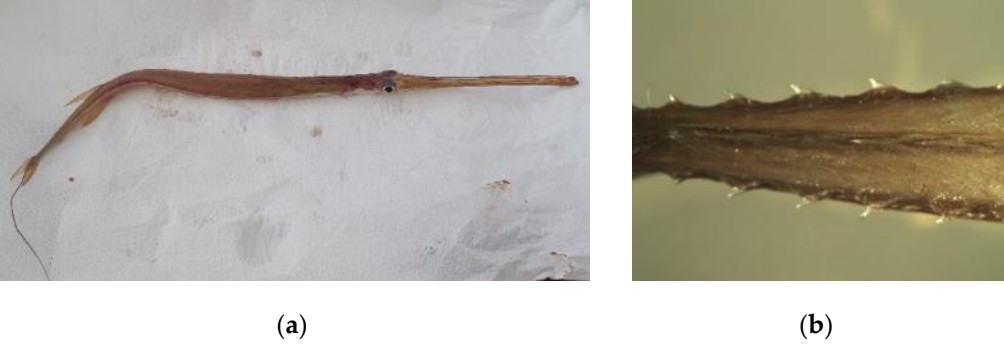

(**a**)                        (**b**)

**Figure 4.** *Fistularia petimba* individual from Amorgos Island and its identification characteristics: (**a**) reddish colour and (**b**) elongated bony plates embedded in the skin.

The morphometric characteristics shown in Table 1 were measured to the nearest second decimal in mm.

**Table 1.** Morphometric characteristics and measurements of *Fistularia petimba* from Amorgos Island, Greece.

| Morphometrics | Measurement (mm) |
|---|---|
| Total Length without filament (TL) | 395.00 |
| Total Length with filament (TLf) | 530.00 |
| Filament Length (fL) | 124.23 |
| Standard Length (SL) | 378.00 |
| Fork Length (FL) | 383.00 |
| Body Deth (BD) | 7.98 |
| Head Length (HL) | 142.00 |
| Eye Diameter (ED) | 10.97 |
| Snout Length (SN) | 114.00 |
| Dorsal Fin Length (DFL) | 15.41 |
| Dorsal Fin Height (DFH) | 28.89 |
| Pectoral Fin Length (PFL) | 6.66 |
| Pelvic Fin Height (PFH) | 16.81 |
| Dorsal Fin Length (PvFL) | 2.54 |
| Pelvic Fin Height (PvFH) | 6.83 |
| Caudal Fin Length (CFL) | 19.01 |
| Caudal Fin Height (CFH) | 5.10 |
| Anal Fin Length (AFL) | 15.16 |
| Anal Fin Height (AFH) | 27.37 |
| Pre-dorsal Fin Length (pDFL) | 67.00 |
| Pre-pectoral Fin Length (pPFL) | 319.00 |
| Pre-pelvic Fin Length (pPvFL) | 190.00 |
| Pre-anal Fin Length (pAFL) | 310.00 |

### 2.3. Distibution of F. petimba in the Mediterranean Sea

To compile a map with records of *F. petimba* in the Mediterranean Sea, a literature review (until February 2024) was performed using Google Scholar. This review used two main keywords, namely "*Fistularia petimba*" and "Red cornetfish", together with additional keywords to assure that *F. petimba* records were not missed: "Mediterranean Sea", "Invasive species", "NNS", "ecology", "habitat", and "lessepsian".

To visualize the species' geographical distribution in the Mediterranean Sea, a map illustrating the occurrences of *F. petimba* was generated using ArcGIS [38] and by integrating data from this study and published records from scientific journals.

### 3. Results

An individual of *F. petimba* (Figure 4) was captured using trammel nets (GTRs) with a mesh size of 26 mm, deployed between 24.5 m and 30.6 m depth on the 27th of May

2023 at 7:50 p.m. and hauled on the 28th of May 2023 at 6:40 a.m. (soak time 11 h and 50 min) off the coast of Katapola Bay, Amorgos Island, Greece (lon = "25.85879922" lat = "36.82760281"). The specimen had a total length of 395.00 mm and a total wet weight of 34 g. The measurements for each of the morphometric characteristics of the species are presented in Table 1.

## 4. Discussion

The increases in global trade and travel have also increased the chances for species to migrate, immigrate, or establish in areas beyond their native ranges, i.e., through widening and deepening canals or ballast water transport [39,40]. Immigration is most commonly associated with human intervention, such as the opening of canals. The Suez Canal, since its completion in the late 19th century, has served as a major conduit for the immigration of marine organisms between the Red Sea (and Indian Ocean) and the Mediterranean Sea. This artificial connection has facilitated the establishment of numerous NNS in the Mediterranean Basin, reshaping the region's biodiversity and ecological dynamics [41,42]. When an area is invaded, it becomes a source for the subsequent spread of the organism to other locations in the basin [41]. Marinas in the Mediterranean Sea have been identified as significant areas for the establishment of NNS, not only for initial introductions, but also for subsequent secondary invasions, acting as stepping stones in the spread of NNS [40,43].

In the marine environment, NNS can become invasive, resulting in the displacement of native species, thus leading to a range of negative consequences. These effects encompass the loss of native genetic diversity, alterations to habitats, shifts in community composition, changes to food web dynamics and ecosystem functions, disruptions to the provision of ecosystem services, threats to human health, and significant economic damages [44].

The capture of *Fistularia petimba* in Amorgos Island (Central Aegean Sea) represents a significant addition to the existing knowledge of the species' distribution in the Mediterranean region. Prior to this finding, 31 sightings were reported from the Mediterranean Sea, out of which only 1 was in the Aegean Sea, specifically in Samos Island (Figure 5, Table 2). The first record in Cadiz, Spain, appears to be an incidental catch from the Atlantic Ocean, while the subsequent occurrences, mainly in the eastern Mediterranean Sea, reveal a progressive invasion pattern from the Indo-Pacific region and the Red Sea into novel habitats. The review of its current distribution reveals its dispersal path along the coasts of Syria, Egypt, and Turkey. The species' course is validated with further records from Cyprus and its possible establishment within the Aegean Sea. Due to the lack of available data on the potential impact of *F. petimba* on native species in the Mediterranean Sea, insights can be drawn from its closely related species, *Fistularia commersonii*. *Fistularia commersonii* is mainly piscivorous but also feeds on crustaceans [32]. It has a high reproductive rate and a prolonged spawning season extending from May to August, allowing for rapid population growth [45]. More than 70% of *Fistularia commersonii*'s diet includes economically valuable species. Its predation on small fish near the seabed, where they hatch and grow, disrupts ecosystem balance and reduces fish biomass [32]. This species quickly establishes itself in new environments, leading to significant ecological and economic consequences, including damage to fisheries (although gradually introduced to consumers) [32,45]. Given the similarities between *F. petimba* and *F. commersonii*, it is likely that *F. petimba* could have similar effects on the Mediterranean ecosystem. Effective monitoring and management strategies are crucial to mitigate these potential impacts. Each documented occurrence represents a critical point in the species' biogeographic spread throughout the eastern Mediterranean Sea. This review of chronological records offers valuable insights into the stepping stone spread of *F. petimba* establishment in the Mediterranean Sea. Tracking its cross-border path from the Red Sea to the eastern Mediterranean Sea with subsequent records in the Aegean Sea provides valuable insights into the species' establishment phase. As reproduction is a crucial part of the successful establishment of species in new areas, Papageorgiou et al. 2023 [46] estimated that the mean total length for gonad maturity was 440 mm for females

and 410 mm for males. The results of Papageorgiou et al. 2023 [46] are in accordance with our results, wherein no visible gonads could be identified macroscopically.

**Table 2.** Data of the 32 validated records of *Fistularia petimba* in the Mediterranean Sea.

| No | Location | Latitude | Longitude | Date (Capture) | Depth (m) | Gear Type | Sample Size |
|----|----------|----------|-----------|----------------|-----------|-----------|-------------|
| 1 | Cadiz, Spain [34] | 36.455097 | −4.703372 | 23 June 1996 | 50 | Gillnet | 1 |
| 2 | Antalya Bay, Turkey [35] | 36.793556 | 31.209167 | 28 October 2016 | 35–43 | Bottom trawl | 1 |
| 3 | Ashdod, Israel [27] | 31.813950 | 34.459717 | 12 November 2016 | 80 | Bottom trawl | 1 |
| 4 | Antalya Bay, Turkey [35] | 36.737417 | 31.434361 | 26 November 2016 | 30 | Bottom trawl | 1 |
| 5 | Iskenderun, Turkey [35] | 36.654400 | 36.186183 | 21 May 2017 | 35–38 | Bottom trawl | 2 |
| 6 | Tripoli, Lebanon [47] | 34.410000 | 35.770000 | 15 November 2017 | N/A | Gillnet | 1 |
| 7 | Mersin Bay, Turkey [37] | 36.128833 | 33.520667 | 22 November 2017 | 95 | Bottom trawl | 1 |
| 8 | Antalya Bay, Turkey [37] | 36.061867 | 32.534233 | 9 January 2018 | 70 | Bottom trawl | 2 |
| 9 | Büyükeceli Coast (Mersin Bay) Turkey [36] | 36.123139 | 33.467944 | 5 October 2018 | 150 | Bottom trawl | 2 |
| 10 | Egypt [28] | El-Hamam—Sidi Kirayr. | | 9 March 2019 | 40–60 | Bottom trawl | 1 |
| 11 | Lattakia, Syria [48] | 35.518325 | 35.713492 | 29 July 2019 | 45 | Gillnet | 1 |
| 12 | Lattakia, Syria [49] | 35.243086 | 35.920000 | 24 September 2019 | 30 | Gillnet | 1 |
| 13 | Gialia, Cyprus [50] | 35.110000 | 32.490000 | 26 September 2019 | 55 | Gillnet | 1 |
| 14 | Banyas, Syria [49] | 35.518325 | 35.713492 | 29 September 2019 | 45 | Gillnet | 2 |
| 15 | Gökova Bay, Turkey [51] | 36.857889 | 27.896556 | 19 October 2019 | 15–20 | Longline | 1 |
| 16 | Güllük Bay, Turkey [51] | 36.857883 | 27.896561 | 17 November 2019 | 65 | Bottom trawl | 4 |
| 17 | Cyprus [46] | 34.747367 | 33.463400 | 14 July 2020 | 55 | Bottom trawl | 3 |
| 18 | Cyprus [46] | 34.964500 | 34.964500 | 15 July 2020 | 48 | Bottom trawl | 1 |
| 19 | Cyprus [46] | 34.759617 | 33.480650 | 16 July 2020 | 33 | Bottom trawl | 1 |
| 20 | Cyprus [46] | 34.924100 | 33.908050 | 24 July 2020 | 79 | Bottom trawl | 11 |
| 21 | Cyprus [46] | 35.081733 | 32.458700 | 24 July 2020 | 43 | Bottom trawl | 29 |
| 22 | Cyprus [46] | 34.635717 | 32.638517 | 27 March 2021 | 46 | Bottom trawl | 10 |
| 23 | Cyprus [46] | 34.661300 | 32.468650 | 27 March 2021 | 93 | Bottom trawl | 4 |
| 24 | Bandirma Bay, Turkey [47] | 40.416950 | 28.084000 | 11 June 2021 | 32 | Trammel net | 1 |
| 25 | Cyprus [46] | 34.693983 | 33.135567 | 4 August 2021 | 44 | Bottom trawl | 26 |
| 26 | Samos, Greece [47] | 37.706583 | 26.708783 | 7 November 2021 | 20 | Trammel net | 1 |
| 27 | Cyprus [46] | 35.060833 | 34.054383 | 8 August 2021 | 86 | Bottom trawl | 4 |
| 28 | Cyprus [46] | 34.750917 | 33.480933 | 8 August 2021 | 55 | Bottom trawl | 7 |
| 29 | Cyprus [46] | 34.750917 | 33.480933 | 8 August 2021 | 33 | Bottom trawl | 2 |
| 30 | Cyprus [46] | 34.692267 | 33.166750 | 8 August 2021 | 56 | Bottom trawl | 1 |
| 31 | Cyprus [46] | 34.699333 | 33.311817 | 13 September 2021 | 13 | Trammel net | 1 |
| 32 | Amorgos, Greece (current study) | 36.82760281 | 25.85879922 | 28 May 2023 | 24.5–30.6 | Trammel net | 1 |

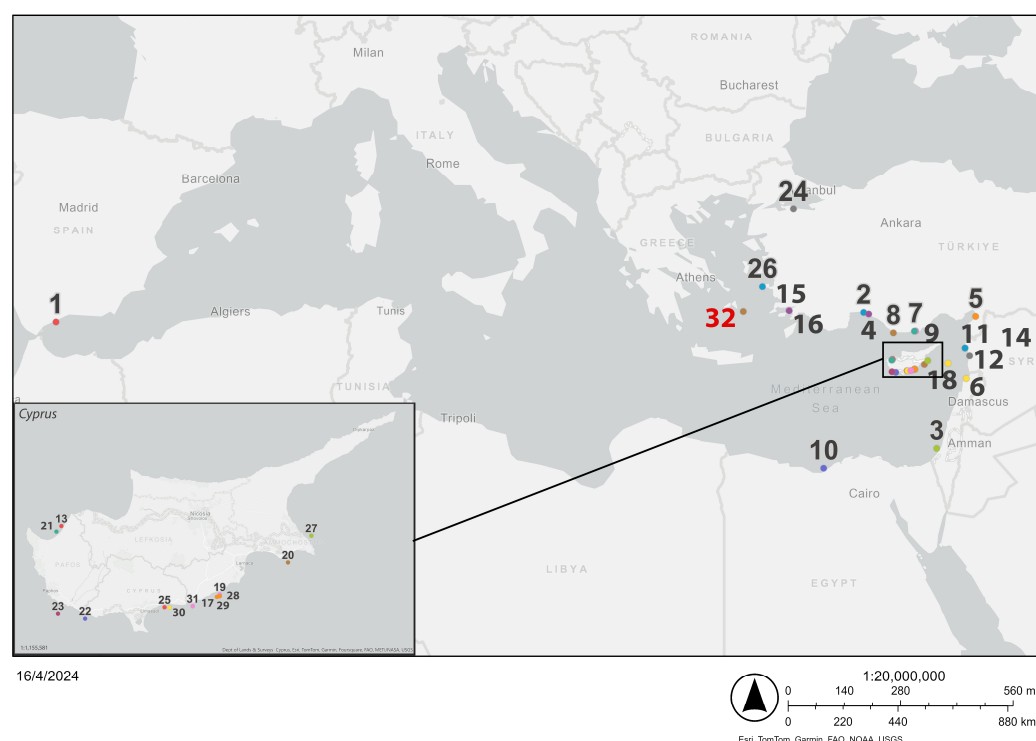

**Figure 5.** Records of *Fistularia petimba* in the Mediterranean Sea (Table 2) [38].

## 5. Conclusions

The occurrence of *F. petimba* in Amorgos Island suggests a wider presence within the Aegean Sea than previously recognized. This finding underscores the importance of ongoing fishery research and marine monitoring in uncovering new occurrences and expanding our understanding of marine biodiversity in the region. Further investigations into the ecological preferences, population dynamics, life traits, and potential impacts of *Fistularia petimba* in the Aegean Sea and the Mediterranean Basin are important to understand succession rates and ecological impacts, to enhance conservation efforts, and to inform sustainable management practices.

**Author Contributions:** Conceptualization, S.K.; Methodology, S.K.; Validation, S.K.; Formal analysis, N.T. and S.K.; Investigation, N.T. and S.K.; Data curation, N.T. and S.K.; Writing—original draft, N.T. and S.K.; Writing—review & editing, N.T. and S.K.; Project administration, S.K.; Funding acquisition, S.K. All authors have read and agreed to the published version of the manuscript.

**Funding:** This research was co-funded by Blue Marine Foundation (BMF) and Cyclades Preservation Fund (CPF)—Conservation Collective (CC) under the project entitled "Base line study for fisheries management on coastal areas based on local ecological knowledge in Amorgos Island".

**Institutional Review Board Statement:** Not applicable for projects funded prior to 2022 (Greek Official Government Gazette Law 4957 FEK A 141/21.07.2022). The project under which data were collected for this study were granted in June 2022; only projects with a starting date after 21 July 2022 are subject to ethical review.

**Informed Consent Statement:** Not applicable.

**Data Availability Statement:** Data is contained within the article.

**Acknowledgments:** The authors thank the Professional Fishing Association of Amorgos and Ioannis Vekris for their expertise and assistance throughout all aspects of our study.

**Conflicts of Interest:** The authors declare no conflicts of interest.

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
