# Peer review of "First Record of the Red Cornetfish Fistularia petimba Lacepède, 1803 from Amorgos Island (Central Aegean Sea; Greece) and a Review of Its Current Distribution in the Mediterranean Sea"

_fishes, doi:10.3390/fishes9060237_

Round 1
Reviewer 1 Report
Comments and Suggestions for Authors
I think this paper is ready to publish when it improves the following subjects and text editing that I suggest:
“F.petimba” must be written with and space between genus and species “F. petimba” in the following lines: 14, 20, 75, 143, 144, 147, 151 and 154.
“Fistularia petimba” must be in italic in the following lines: 186, 191, 198, 304, 306, 317, 320, 322, 325, 327, 362, 364, 369 and 371.
In line 45, the authors should explain what they mean by “specific habitats”.
When it refers to the genus Fistularia, it should be written in italic: lines 56 and 108.
F.tabacaria” must be written with and space between genus and species “F. tabacaria”: line 58.
Figure 1 should be improved. Those of us who are not familiar with the area are unable to locate the study area. A scale is needed. The authors must explain in the text why they mention FRAs in the figure caption. What does it have to do with the study?
The units of the fishing gears must be specified in lines 89 to 98.
In line 108 Fritzsche (1976) mustn’t be written in italic.
In line 129 and Table 1 I assume the authors are referring to “Body Depth” instead of “Body Deth”.
The quality of figure 5 must be improven. It's impossible to read the toponyms of the enlarged figure.
In the discussion, citation numbers should be expressed as superscripts.
Author Response
Comment 1: “F.petimba” must be written with and space between genus and species “F. petimba” in the following lines: 14, 20, 75, 143, 144, 147, 151 and 154.
Response 1: Revised as suggested.
Comment 2: “Fistularia petimba” must be in italic in the following lines: 186, 191, 198, 304, 306, 317, 320, 322, 325, 327, 362, 364, 369 and 371.
Response 2: Revised as suggested.
Comment 3: In line 45, the authors should explain what they mean by “specific habitats”.
Response 3: The term “specific habitats” is extensively covered in the referenced literature, discussing various habitat types and non-native species. However, for the purpose of this study, we believe additional explanation in this context is unnecessary.
Comment 4: When it refers to the genus Fistularia, it should be written in italic: lines 56 and 108.
Response 4: Revised as suggested.
Comment 5: “F.tabacaria” must be written with and space between genus and species “F. tabacaria”: line 58.
Response 5: Revised as suggested.
Comment 6: Figure 1 should be improved. Those of us who are not familiar with the area are unable to locate the study area. A scale is needed.
Response 6: Revised to 300 dpi.
Comment 7: The authors must explain in the text why they mention FRAs in the figure caption. What does it have to do with the study?
Response 7: The term was removed as suggested. FRAs were a part of the study in Amorgos Island but there is no connection in the specific paper.
Comment 8: The units of the fishing gears must be specified in lines 89 to 98.
Response 8: The units of mesh size in nets are mm and the information is added to the text. The hook sizes do not have a unit and the numbers 9, 10, 11, 12, 13, 14 refer to the commercial sizes.
Comment 9: In line 108 Fritzsche (1976) mustn’t be written in italic.
Response 9: Revised as suggested.
Comment 10: In line 129 and Table 1 I assume the authors are referring to “Body Depth” instead of “Body Deth”.
Response 10: The paragraph was removed as suggested by Reviewer 2
Comment 11: The quality of figure 5 must be improven. It's impossible to read the toponyms of the enlarged figure.
Response 11: Revised to 600 dpi.
Comment 12: In the discussion, citation numbers should be expressed as superscripts.
Response 12: Revised as the editors suggested, all references should be place
the numbers in square brackets (“[ ]”), e.g., [1], [1–3], or [1,3].
Reviewer 2 Report
Comments and Suggestions for Authors
Unfortunately, I have to recommend that the MS in its current state be rejected. There are two major problems with it:
1) The English is very poor. In addition, a large section of the Methods (lines 42-52) would be best placed in Results of Discussion. You have extensive abbreviations yet repeat them in Table 1. Repetition like this is a waste of space. Was the specimen deposited into a collection? If so, where and what is its registration number? Figure 1 I can hardly make out the red and yellow.
2) The point of the paper, a new record, is just not worthy of the fuss made about it. Looking at the distribution (Figure 5), it is just another dot in a cluster of dots. Given that Fistularia petimba is nearly global in distribution, and that the authors record it benign in the Mediterranean from Cadiz to Tripoli, Egypt to Bandirema Bay, there is nothing significant in it now being discovered off Amorgos Island, just across the water from Samos.
A better focus would be to look at the impact on native fishes and invertebrates by Fistularia petimba.
Comments on the Quality of English LanguageThe English appears to have been translated by AI. Sentences are long and convoluted, and a large number are extraneous fluff and should be deleted. My initial sweep through removed nearly one-third of what was written.
The legends for Figure 5 & Table 2; 'do you mean Citings' (= previously recorded) or 'Sightings'? If the former, what published records?
Table 2, the date of 'capture', were the specimens caught, observed or recorded (caught & discarded/sold). The information as presented is ambiguous.
Author Response
Comment 1: The English is very poor.
Response 1: While we appreciate the feedback on the language quality, we've made some adjustments to improve clarity without extensive alterations. We remain open to further suggestions for improvement.
Comment 2: In addition, a large section of the Methods (lines 42-52) would be best placed in Results of Discussion.
Response 2: Thank you for your suggestion. However, we believe that the section in question is best placed in the Introduction to establish the context and framework for the study. It serves to connect the paragraphs seamlessly and provide background information. Nonetheless, if the editors determine that it would be more suitable in the Results or Discussion section, we are open to their decision.
Comment 3: You have extensive abbreviations yet repeat them in Table 1. Repetition like this is a waste of space.
Response 3: The paragraph was removed as suggested.
Comment 4: Was the specimen deposited into a collection? If so, where and what is its registration number?
Response 4: The specimen is stored at the Agricultural University of Athens (line 103) along with all the specimens of the study in Amorgos Island. A minor adjustment has been made to enhance clarity in the text. We can provide the registration number from our database if needed.
Comment 5: Figure 1 I can hardly make out the red and yellow.
Response 5: Revised to 300 dpi.
Comment 6: The point of the paper, a new record, is just not worthy of the fuss made about it. Looking at the distribution (Figure 5), it is just another dot in a cluster of dots. Given that Fistularia petimba is nearly global in distribution, and that the authors record it benign in the Mediterranean from Cadiz to Tripoli, Egypt to Bandirema Bay, there is nothing significant in it now being discovered off Amorgos Island, just across the water from Samos.
Response 6: Respectfully, we appreciate your perspective but it is our belief that the new record holds significance. The capture off Amorgos Island, despite its proximity to other records, enriches our knowledge of the species' presence in the region. Moreover, considering the ecological implications and potential impacts of its distribution, we believe this finding warrants attention and discussion in the scientific community.
Comment 7: A better focus would be to look at the impact on native fishes and invertebrates by Fistularia petimba.
Response 7: Thank you for your suggestion. However, due to the lack of available data for Fistularia petimba, including an analysis of its impact on native fishes and invertebrates would require extensive research and could significantly lengthen the paper. We could consider elaborating on the impact of the relative species Fistularia commersonii, but this would need further investigation. Ultimately, I would value the opinion of the editors on how to best balance the scope and length of the paper.
Comment 8: The English appears to have been translated by AI. Sentences are long and convoluted, and a large number are extraneous fluff and should be deleted. My initial sweep through removed nearly one-third of what was written.
Response 8: Thank you for your feedback. We have made some alterations to address the issues you raised.
Comment 9: The legends for Figure 5 & Table 2; 'do you mean Citings' (= previously recorded) or 'Sightings'? If the former, what published records?
Response 9: The term “citings” was revised to “records” as suggested. The published records are found in the literature.
Comment 10: Table 2, the date of 'capture', were the specimens caught, observed or recorded (caught & discarded/sold). The information as presented is ambiguous.
Response 10: “Capture Date” refers to the date caught, observed and recorded, the samples were not discarded or sold. So, there was no change made.
Reviewer 3 Report
Comments and Suggestions for Authors
This article reports about finding of species Fistularia petimba near Amorgos Island and it reviews its current distribution from literature. The accent is on the non-native species, especially Lessepsian species that have entered the Mediterranean Sea after building the Suez canal. This manuscript is informative and well written but some parts are unnecessarily extensive. For example it is not necessary to write about F. petimba morphological characteristics since it was already written a long ago (reference 31 from ref list). I propose that this paper could be shortened and written as a short note (short report) or possibly empowered by assessing the invasive potential of this non-native species (using tools such as AS-ISK or similar).
Please use non-native instead of non-indigenous. In recent studies (Soto et al, 2024) (https://www.researchgate.net/publication/378538375_Taming_the_terminological_tempest_in_invasion_science?enrichId=rgreq-f62a910ba10c101fb6cb822f419ad70e-XXX&enrichSource=Y292ZXJQYWdlOzM3ODUzODM3NTtBUzoxMTQzMTI4MTIzMDE4MzU5MUAxNzEwODM0NDI3NTg1&el=1_x_3&_esc=publicationCoverPdf ) about terminology it is said that “Consequently, politically charged terms like ‘invasive’ or colonial terms such as ‘non-indigenous’, ‘naturalised’, or ‘colonised’ can be circumvented.“
Line 22. Please provide keywords that are different from those in title
Sentence in lines 39, 40 and 41:
“Additionally, recent findings during the past decades, points Mediterranean ports as a major hotspot for the introduction of NIS in this basin” is not connected with reference no. 19 (Romeo et al, 2015: First evidence of presence of plastic debris in stomach of large pelagic fish in the Mediterranean Sea). This paper reports about plastic ingestion of large pelagic fish. Please remove or replace reference 19.
Line 103. All samples of which species, F. petimba or all species? This is a little confusing, I understood it was only one specimen.
Line 107. I don’t find mentioning this species characteristic necessary, since in literature you can find these data already (REF 31 in your reference list)
Line 138-142. In the results it is stated that the stomach was empty and sex couldn’t be determined, so this part in M&M wouldn't be necessary.
Line 164, 165. Table 1, measurement list: similar as for line 107, these measures are not necessary and it is a little bit confusing having the same measures in mm and g. You could mention the total length and weight of species in text.
Line 166. Please write F. petimba in italic in discussion
Line 168, 175, 176, 179, 185. References number not in exponent
Line 206. Please indicate in the figure caption what is the name of the island (Cyprus) that is zoomed in on the map and that the numbers could be found in the table.
Author Response
Comment 1: This article reports about finding of species Fistularia petimba near Amorgos Island and it reviews its current distribution from literature. The accent is on the non-native species, especially Lessepsian species that have entered the Mediterranean Sea after building the Suez canal. This manuscript is informative and well written but some parts are unnecessarily extensive. For example it is not necessary to write about F. petimba morphological characteristics since it was already written a long ago (reference 31 from ref list).
Response 1: The paragraph was removed as suggested.
Comment 2: I propose that this paper could be shortened and written as a short note (short report) or possibly empowered by assessing the invasive potential of this non-native species (using tools such as AS-ISK or similar).
Response 2: Thank you for the suggestion. I appreciate the idea of shortening the paper to a short note. However, expanding the scope to assess the invasive potential using tools like AS-ISK would require additional data and extend the length of the manuscript. I would value the opinion of the editors on how to best proceed.
Comment 3: Please use non-native instead of non-indigenous. In recent studies (Soto et al, 2024) (https://www.researchgate.net/publication/378538375_Taming_the_terminological_tempest_in_invasion_science?enrichId=rgreq-f62a910ba10c101fb6cb822f419ad70e-XXX&enrichSource=Y292ZXJQYWdlOzM3ODUzODM3NTtBUzoxMTQzMTI4MTIzMDE4MzU5MUAxNzEwODM0NDI3NTg1&el=1_x_3&_esc=publicationCoverPdf ) about terminology it is said that “Consequently, politically charged terms like ‘invasive’ or colonial terms such as ‘non-indigenous’, ‘naturalised’, or ‘colonised’ can be circumvented.“
Response 3: The term non-indigenous was revised to non-native as suggested.
Comment 4: Line 22. Please provide keywords that are different from those in title.
Response 4: Thank you for your feedback. We revised the keywords and added “Marine biodiversity” and “Species distribution”.
Comment 5: Sentence in lines 39, 40 and 41: “Additionally, recent findings during the past decades, points Mediterranean ports as a major hotspot for the introduction of NIS in this basin” is not connected with reference no. 19 (Romeo et al, 2015: First evidence of presence of plastic debris in stomach of large pelagic fish in the Mediterranean Sea). This paper reports about plastic ingestion of large pelagic fish. Please remove or replace reference 19.
Response 5: The reference was revised as suggested. The correct reference (Zenetos, A.; Ovalis, P.; Giakoumi, S.; Kontadakis, C.; Lefkaditou, E.; Mpazios, G.; Simboura, N.; Tsiamis, K. Saronikos Gulf: a hotspot area for alien species in the Mediterranean Sea. BioInvasions Records 2020, 9 (4), 873–889. DOI: https://doi.org/10.3391/bir. 2020.9.4.21) was added.
Comment 6: Line 103. All samples of which species, F. petimba or all species? This is a little confusing, I understood it was only one specimen.
Response 6: Revised as suggested.
Comment 7: Line 107. I don’t find mentioning this species characteristic necessary, since in literature you can find these data already (REF 31 in your reference list)
Response 7: The paragraph was removed as suggested.
Comment 8: Line 138-142. In the results it is stated that the stomach was empty and sex couldn’t be determined, so this part in M&M wouldn't be necessary.
Response 8: The paragraph was removed as suggested.
Comment 9: Line 164, 165. Table 1, measurement list: similar as for line 107, these measures are not necessary and it is a little bit confusing having the same measures in mm and g. You could mention the total length and weight of species in text.
Response 9: Revised as suggested.
Comment 10: Line 166. Please write F. petimba in italic in discussion.
Response 10: Revised as suggested.
Comment 11: Line 168, 175, 176, 179, 185. References number not in exponent
Response 11: Revised as the editors suggested, all references should be place
the numbers in square brackets (“[ ]”), e.g., [1], [1–3], or [1,3].
Comment 12: Line 206. Please indicate in the figure caption what is the name of the island (Cyprus) that is zoomed in on the map and that the numbers could be found in the table.
Response 12: Revised as suggested.
Reviewer 4 Report
Comments and Suggestions for Authors
This MS can be accepted for publication, although the authors were honored to create such an extensive text on such a particular finding. Nevertheles, it is scientifically sound. I only recommend assessing the novelty of this finding more modestly along the text, since the distance from the previously known capture sites is small.
Figures 1 and 5 should be united and the geographical names should be signed clearly.
Author Response
Comment 1: I only recommend assessing the novelty of this finding more modestly along the text, since the distance from the previously known capture sites is small.
Response 1: Thank you for your suggestion. We respectfully maintain our assessment of the novelty of our finding. We believe it contributes significantly to the field despite the proximity to known capture sites.
Comment 2: Figures 1 and 5 should be united and the geographical names should be signed clearly.
Response 2: We intentionally kept Figures 1 and 5 separate for clarity. Figure 1 depicts the sampling area, while Figure 5 compiles all records of Fistularia petimba in the Mediterranean Sea. Combining them into a single map might be confusing. We decided not to make the suggested change because darkening the geographical names on Figure 5 might make it harder to read the overlapping names and the number of records, causing more confusion.
Round 2
Reviewer 2 Report
Comments and Suggestions for Authors
Yes, include the registration number of the voucher specimen so that in decades to come, the same specimen can be re-examined in the light of your paper.
Comments on the Quality of English LanguageThe English and punctuation still need some minor adjusting (e.g. lines 45-52). However, I just don't have the time.
Reviewer 3 Report
Comments and Suggestions for Authors
Dear authors and editor,
I believe that the paper in its current form has been sufficiently revised and improved to be published in your journal. It is up to the editor to decide whether it will be in the form of a short communication or a full paper.
All the best
Author Response
Thank you!